# Exploring Sedentary and Nutritional Behaviour Patterns in Relation to Overweight and Obesity Among Youth from Different Demographic Backgrounds in Saudi Arabia

**DOI:** 10.3390/ijerph22050813

**Published:** 2025-05-21

**Authors:** Anwar Al-Nuaim, Ayazullah Safi

**Affiliations:** 1Physical Education Department, Education College, King Faisal University, Al-Ahsa 31982, Saudi Arabia; 2Department of Public Health, College of Life Sciences, Birmingham City University, Birmingham B15 3TN, UK; ayazullah.safi@bcu.ac.uk

**Keywords:** lifestyle, obesity, overweight, sedentary, nutrition, youth, Saudi Arabia

## Abstract

Background: The prevalence of overweight and obesity has increased over the last three decades, becoming a major public health concern. This issue is particularly pressing in terms of the impact it has on the population of the Kingdom of Saudi Arabia (KSA). Thus, the aim of this research was to explore the nutritional and lifestyle habits of youth in the Al-Ahsa region of the KSA. Methods: This cross-sectional study included a total of 1270 secondary-school boys and girls (15–19 years) from all five of the Al-Ahsa Governorate districts. BMI and waist circumference measurements were recorded using anthropometric measurements, and the lifestyle–Arab Teens Lifestyle Survey (ATLS) was used to measure sedentary and eating habits. Results: Chi-square analysis revealed that a higher proportion of females (90.68%) than males (79.18%) exceeded two hours of daily sedentary time. Frequent sugary drink consumption (>3 times per week) was similar in males (67.40%) and females (66.11%). Students from different geographical locations showed significant differences in exceeding cut-off scores for fast food (χ^2^ = 24.503, *p* < 0.001), cakes/doughnuts (χ^2^ = 8.414, *p* = 0.015), sweets/candy (χ^2^ = 19.613, *p* < 0.001), and energy drinks (χ^2^ = 21.650, *p* < 0.001). Conclusions: Al-Ahsa is the largest governorate in the KSA. It has some of the largest health risks regarding obesity and poor lifestyle habits. This study highlights the major need for future research and policy interventions.

## 1. Background

A growing body of evidence indicates that the prevalence of overweight and obesity has increased over the last three decades, becoming a major public health concern in both developed and developing countries [1,2,3]. The health-related risks of obesity are well established, including hypertension, type II diabetes, and coronary heart disease, as well as psychological problems, such as depression and low self-esteem [3,4]. It is well documented that obesity occurs when energy intake exceeds the expenditure [4]. Obesity is reported to be the fifth leading risk factor for global mortality [5]. In addition, research shows that obesity has been found to track from childhood to adulthood [4,6]. Furthermore, studies has shown that the incidence of obesity among youth and adult populations in the Kingdom of Saudi Arabia (KSA) is higher compared to Western Europe and the USA [7]. Furthermore, a national survey conducted between 1995 and 2000 found that the prevalence of obesity among the population of Saudi Arabia was 35.6% [8]. Some of the key contributing factors include a poor diet, a sedentary lifestyle, the expansion of urbanisation, modes of transportation, the availability of entertainment gadgets, and lack of physical activity (PA) engagement [9,10,11]. Furthermore, Al-Nuaim and Safi [12] conducted a study focused on the correlation between built environments, hypertension and weight status among youth in the KSA, the results showed that boys were spending more time being sedentary (3.14) compared to girls (2.96), measured in hours per day, and significant differences were found (F1,379 = 16.50, *p* < 0.001) between boys (mean = 75.80 cm) and girls (mean = 70.38 cm) regarding waist circumference.

Over the past three decades, the Gulf Cooperation Council (GCC), including the KSA, has witnessed substantial lifestyle changes due to economic expansion. In the KSA, the traditional diet of fibre-rich and low-fat has changed to a more Westernised diet, with a higher content of sugar, sodium, cholesterol, and fat. In addition, as the Saudi landscape has become more urbanised, this has resulted in the dominance of motorised transport. Such drastic changes to the physical environment had major health and social implications for the KSA population. Furthermore, the increased availability of satellite television, internet, and video games has also contributed to declining PA and increased sedentary behaviour. The rapid transformation of KSA lifestyle and culture is thought to have played a major role in the prevalence of non-communicable diseases reaching epidemic proportions within Gulf region [11,12,13,14]. Furthermore, research suggests that the frequency of obesity is particularly high within the Eastern province of the KSA [14,15,16]. Despite this, there is scarce literature on the occurrence of obesity related to health and lifestyle factors among youth from the Eastern province of KSA [11,12,14], with majority of previous studies focusing on the central province of KSA. Given the significant impact of obesity on population of the KSA, it is important to focus on the relatively under-researched health and lifestyle behaviours of young people in the country. Therefore, the aim of this research was to explore the nutritional and lifestyle habits of youth in the Al-Ahsa region of the KSA.

## 2. Methods

### 2.1. Participants and Procedure

This study was conducted in accordance with the Declaration of Helsinki, after gaining an ethical approval from the Institutional Ethical Review Board of King Faisal University (KFUREC-2022-APR-EA000564). A total of 1270 secondary-school boys and girls (15–19 years) living in the Al-Ahsa Governorate participated, following the distribution of informed participant and parental consent forms. The primary objective of the study was to provide reliable population estimates of sedentary and nutritional patterns and health habits. The sample design called for wide geographical coverage of Al-Ahsa school communities (i.e., urban, rural farm, and rural desert “Bedouin”), with representation of populations in urban, rural, state, and private secondary schools. The sample size was determined to be within (±0.05) of the population proportion with a 95% confidence level. This study was cross-sectional in design and included participants from all five of the Al-Ahsa Governorate districts, namely Al-Hofuf, Al-Mubaraz, East Villages, North Villages, and Al-Hejar. A secondary school (one all-boys and one all-girls) from each district was randomly selected to take part in this study. Also, two private schools for each gender were randomly selected from the Al-Hofuf and Al-Mubaraz districts. In total, seven schools were selected to participate in this study, representing urban, rural, and rural farm geographical locations. The stratified sample, representing different geographical areas of the Governorate, included 607 females (*M =* 17.07 ± 1.27 yrs) and 663 males (*M* = 17.08 ± 1.10 yrs).

### 2.2. Measures

#### Anthropometric Measurements

In order to objectively assess weight status classification and the prevalence of overweight and obesity among youth, BMI and waist circumference measurements were recorded. Body weight was measured to the nearest 100 g using Seca weight scales (Seca Ltd., Hamburg, Germany). The height was measured to the nearest 0.5 centimetre using a Seca portable height measure (Seca Ltd., Hamburg, Germany). BMI was calculated using the formula: weight (kg)/height (m^2^). BMI was classified according to the International Obesity Task Force (IOTF) criteria [17]. The waist circumference was obtained to the nearest 0.5 cm as a cut-off point [18]. Those who exceeded 0.5 cm were considered at risk of cardiovascular diseases. A non-stretchable measuring tape (Richter measuring tape, Seca Ltd., Hamburg, Germany) was used to measure waist circumference horizontally to the nearest 0.5 cm, at the level of umbilical and at the end of normal expiration. The cut-off point for waist circumference classification was a waist-to-height ratio of 0.5 cm, with youth who exceed 0.5 cm being considered “at risk” of cardiovascular diseases [19]. Waist circumference was utilised in addition to BMI, as it is considered a simple measure of fat distribution in youth and is least affected by gender, race, and overall adiposity [20].

The lifestyle–Arab Teens Lifestyle Survey (ATLS) was used to measure sedentary and eating habits. Previous research reported that ATLS is valid and reliable and has been used by previous research among KSA youth [16,21,22]. The ATLS wasdetermined to measure SB as time typically spent on daily activities including watching TV, playing video games, and Internet use. In accordance with previous research, spending more than two hours continually on screen, including internet, playing computer games or watching TV, was considered as sedentary; thus, cut-off point was applied to this study. The dietary habits measurement included questions related to how many times per typical week participants consumed breakfast, sugar-sweetened drinks including soft beverages, vegetables (cooked and uncooked), fruit, milk and dairy products, doughnuts and cakes, candy and chocolate, energy drinks, and fast foods. The fast foods in this regard included examples from both Western fast foods and Arabic fast-food choices, such as shawarma (grilled meat or chicken in pitta bread with some salad). These questions covered healthy and unhealthy dietary habits. The students had a choice of answers, ranging from zero intake to a maximum intake of seven days per week (every day). As conducted in a previous study [21], for the dietary cut-off points, the percentage of adolescents who had a daily intake of breakfast, fruit, vegetables and milk was calculated for healthy dietary habits, whilst those who exceeded three days’ intake per week of fast food or sugary drinks were classified as having unhealthy dietary habits.

### 2.3. Statistical Analysis

A range of statistical procedures were performed using SPSS, version 24 to establish associations and differences in lifestyle habits of youth from different locations and genders within Al-Ahsa Governorate. In addition, comparisons between genders, geographical locations, and age groups were conducted using 2-way and 3-way analyses of variance (ANOVA), BMI, and waist circumferences. A two-step cluster analysis was used to identify different risk factors based on dietary habits. This analysis was used to identify subgroups of cases in specific populations based on shared characteristics. Subsequent Chi-square and analysis of variance (ANOVA) were used to identify differences between the clusters regarding geographical locations and demographic characteristics. Also, analysis were used to identify differences (based on nutrition habits) in weight status and SB. Furthermore, Pearson’s correlations were performed to establish relationships between health status variables such as BMI, waist circumference, nutrition and SB habits, including watching TV time and computer usage.

## 3. Results

The descriptive characteristics of the main dependent variables for the total sample and sub-samples are presented in Table 1. There were no significant differences between males and females in TV viewing or computer usage. When SBs were combined, Chi-square analysis indicated that the proportion of youth exceeding two hours per day being sedentary during their free time was significantly higher in females than males (90.68% and 79.18%, respectively). This shows less than 10% of females and approximately 21% of males reached the recommended guideline of less than two hours per day. With regard to gegraphical location, the proportion of youth who exceeded two hours per day of sedentary time was significantly higher in urban youth (87.8%) compared to youth from rural farms (82.4%) or rural desert (77.6%). In addition, the proportion of rural desert youth (30%) who exceeded two hours per day on the computer was significantly lower than urban (49.3%) or rural farm youth (44.4%).

There was no significant difference between age groups in sedentary time; however, there was a significant difference between state and private schools in screen time (*p* = 0.003). For instance, students from private school, on average, were spending more hours watching TV or using a computer (5.87 h per day) than students from state schools (5.23 h per day). The Chi-square analysis indicated that there was no significant difference between the percentage of students who exceeded the recommended guideline of two hours or less per day (87.77%) from state school and (87.73%) from private school students spending more than two hours watching TV or using the computer.

### 3.1. Differences in Dietary Habits

There was a statistically significant difference between gender by location interaction for most of the dietary habits selected. In general, a higher proportion of males compared to females exceeded the cut-off scores for most of the dietary habits, including daily consumption of breakfast, fruits, vegetables, milk and dairy products, and fast food. On the other hand, a higher proportion of females than males exceeded the cut-off scores of more than three days of consumption per week of chips and crisps, cakes and doughnuts, and sweets and chocolate. Table 2 illustrates that around two-thirds of males (67.40%) and females (66.11%) consumed sugary drinks more than three times per week. In addition, whilst males consumed significantly more healthy foods compared to females, the majority of both males and females did not consume healthy foods daily, including breakfast (76.84% and 83.31%), vegetables (74.96% and 80.8%), fruit (84.57% and 90.79%), and milk and dairy intake (70.93% and 78.26%). Moreover, around two-thirds of females consumed chips/crisps (63.48%) or sweets/chocolate (62.10%) on more than three days per week, whereas the proportion of males eating these foods on more than three days per week was significantly lower (30.39% and 38.11%).

#### 3.1.1. Inferential Nutritional Statistics Across Age

Univariate ANOVA revealed there was no significant difference between age groups in most nutritional habits; this includes breakfast, milk, vegetables, fruits, sugary drinks, fast food, chips/crisps and energy drinks (*p* > 0.05). However, there were significant differences between age groups in consumption of cakes/doughnuts (*p* = 0.003) and sweets/candy (*p* < 0.001). Fifteen-year-olds consumed significantly more cakes/doughnuts (3.57 times per week) than 19-year-old students (2.68 times per week). Furthermore, there was no significant difference between other age groups. In addition, 15-year-olds ate more sweets/candy (4.60 times per week) than 18-year-old students (3.49 times per day) and 19-year-old students (3.15 times per day). Moreover, 17-year-olds ate sweets/candy (3.90 times per week) significantly more often than 19-year-old students (3.15 times per day). The Chi-square analysis indicated there was a significant difference between age groups in sweets/candy consumption (*χ*^2^_4_ = 17.545, *p* = 0.002). The highest percentage of students who exceeded sweets/candy cut-off scores was 15-year-olds (64.28%), whereas 19-year-olds had the lowest percentage (37.80%).

#### 3.1.2. Inferential Nutritional Statistics Across Geographical Locations

Univariate ANOVA revealed significant differences between geographical locations in fast food (*p* < 0.001), cakes/doughnuts (*p* = 0.005), sweets/candy (*p* < 0.001) and energy drinks consumption (*p* < 0.001). Urban youth ate fast food (3.15 times per week) significantly more than those on rural farms (2.64 times per week) and in rural desert (2.08 times per week). Moreover, youth from rural farms ate fast food significantly more compared to rural desert youth (*p* = 0.007). In addition, rural desert youth ate cakes/doughnuts (3.56 times per week) significantly more than youth from urban (*p* = 0.009, 2.98 times per week) or rural farm students (*p* = 0.006, 2.91 times per week). However, there was no significant difference between urban youth and rural farm youth in cakes/doughnuts consumption (*p* > 0.05). Moreover, rural farm youth ate sweets/candy (3.29 times per week) significantly less than youth from urban (*p* < 0.001, 4.01 times per week) and rural desert areas (*p* = 0.049, 3.82 times per week); however, there was no significant difference between urban and rural desert youth in sweets/candy consumption. Finally, rural farm youth drank energy drinks (0.81 times per week) significantly less than urban youth (*p* < 0.001, 1.60 times per week) or rural desert youth (*p* = 0.005, 1.45 times per week). However, there was no significant difference between urban and rural desert youth in energy drinks consumption (*p* > 0.05).

Chi-square analysis indicated that there was a significant difference between students from different geographical locations who exceeded the cut-off scores for fast food (*χ*^2^_2_ = 24.503, *p* < 0.001), cakes/doughnuts (*χ*^2^_2_ = 8.414, *p* = 0.015), sweets/candy (*χ*^2^_2_ = 19.613, *p* < 0.001) and energy drink consumption (*χ*^2^_2_ = 21.650, *p* < 0.001). Urban youth had the highest percentage of students who ate fast food more than three times per week (35.57%) compared to 16.87% of rural desert youth and 27.6% of rural farm youth. In cakes/doughnuts consumption, rural desert youth had the highest percentage of exceeding the cut-off scores at 45.63%, whereas urban and rural farm students had 34.98% and 32.81%, respectively. In addition, rural farm youth had the lowest percentage of students who exceeded the cut-off scores of sweets/candies (40.46%), whereas urban and rural desert youth had similar percentages (54.23% and 53.13%, respectively). Finally, the urban area had the highest percentage of students who exceeded the cut-off scores of energy drinks consumption (19.51%), whereas students from rural farm areas had the lowest percentage (8.76%) and rural desert students had 16.25%. However, there was no significant difference between geographical location in breakfast, soft drink, milk, vegetable, and fruit consumption.

#### 3.1.3. Inferential Nutritional Statistics Across State and Private Schools

Statistical analysis indicated that students from state schools consumed more breakfasts, vegetables, fruits, cakes/doughnuts, and sweets/candy compared to students from private schools. However, students from private schools consumed chips/crisps and energy drinks more than state students. For example, univariate ANOVA revealed that there was a significant difference between state and private schools’ breakfast (*p* = 0.031), vegetables (*p* = 0.025), fruit (*p* = 0.002), chips/crisps (*p* < 0.001), cakes/doughnuts (*p* = 0.006), sweets/candy (*p* = 0.029) and energy drinks consumption (*p* < 0.001). Students from state schools consumed more than those from private schools in breakfast at home (3.16 times per week, 2.73 times per week, respectively), vegetables (3.76 times per week, 3.33 times per week, respectively), fruit (3.22 times per week, 2.69 times per week, respectively), cakes/doughnuts (3.18 times per week, 2.72 times per week, respectively) and sweets/candy (4.17 times per week, 3.79 times per week, respectively), whereas private school students consumed more than their counterparts did in state school in chips/crisps (4.06 times per week, 3.23 times per week, respectively) and energy drinks (2.11 times per week, 1.27 times per week, respectively). Furthermore, Chi-square analysis revealed that state school students significantly exceeded the cut-off scores more than three days consumption per week of chips and crisps (53.94%, 40.27% respectively) and cakes/doughnuts (53.18%, 29.35% respectively), whereas students from private school, significantly more than state students, exceeded the cut-off scores of more than three days consumption per week of fast food (39.93%, 32.32%, respectively) and energy drinks (25.93%, 14.72%, respectively). Moreover, the students from private schools had significantly lower percentage eating daily breakfast (*χ*^2^_2_ = 6.885, *p* = 0.009). It was found that 85.37% of students from private schools did not eat breakfast daily at home, compared to 77.41% of youth in state schools.

Data on nutritional habits and SB are displayed in Table 3. This includes the mean and standard deviation as well as the proportion of adolescents exceeding the specific cut-off values for nutrition variables and screen time.

Among males, a Binary Logistic Regression analysis revealed that for every hour that males spent watching TV, the risk of being overweight increased by 11% (OR = 1.112; 95% CI1.011–1.223). In addition, for every day that breakfast is missed, males are more likely to be overweight by 18% (OR = 0.926; 95% CI 0.863–0.994). Moreover, for every time per week males consumed dairy products, the risk of becoming overweight increased by 12% (OR = 1.124; 95% CI1.041–1.214). Furthermore, every time per week males consumed chips, the risk of becoming overweight decreased by 21% (OR = 0.894; 95% CI 0.822–0.974). Finally, for every energy drink consumed per week, males were more likely to be overweight by 10% (OR = 1.106; 95% CI 1.021–1.198) (see Table 4). Among females, for every day breakfast was missed, females were more likely to be overweight by 18% (OR = 0.922; 95% CI 0.859–0.989). Moreover, for every energy drink consumed per week, female youth were more likely to be overweight by 20% (OR = 1.209; 95% CI 1.117–1.308) (Table 5).

Positive relationships between BMI were found with some lifestyle factors: computer time used (*r* = 0.077, *p* = 0.007), sedentary time (*r* = 0.080, *p* = 0.005), and energy drink (*r* = 0.165, *p* < 0.001). The intake of breakfast (*r* = −0.103, *p* < 0.001), and chips/crisps (*r* = −0.056, *p* = 0.050).

#### 3.1.4. Cluster Analysis

The following cluster analysis is based on the emergence of two cluster groups: healthy and unhealthy habits. Whilst further analysis was conducted to uncover more potential subgroups, the subsequent findings did not reveal any further identifiable clusters.

Cluster 1 (unhealthy habits group): This cluster was shaped by an average of sweet candy (5.66 times/week), chips/crisps (5.42 times/week), cakes/doughnuts (4.63 times/week), sugary drink (6.20 times/week), fast food (3.79 times/week), energy drink (2.18 times/week), breakfast (2.54 times/week), milk (3.68 times/week), vegetables (3.39 times/week) and fruit (2.81 times/week).

Cluster 2 (healthy habits group): This cluster was shaped by an average of sweet candy (2.53 times/week), chips/crisps (2.42 times/week), cakes/doughnuts (2.01 times/week), sugary drink (3.94 times/week), fast food (2.25 times/week), energy drink (0.79 times/week), breakfast (3.37 times/week), milk (3.92 times/week), vegetables (3.61 times/week) and fruit (3 times/week).

Chi-square analysis indicated that the percentage of cluster 1 (unhealthy group) was significantly (*χ*^2^_1_ 75.341, *p* < 0.001) higher in females (51.61%) than males (27.40%). Regarding the geographical location, Chi-square analysis indicated that there was significant (*χ*^2^_2_ = 12.530, *p* = 0.002) differences between school types (i.e., urban, rural farm and rural desert), with the highest proportion of cluster 1 (unhealthy group) among urban youth (42.85%) and the lowest proportion among rural farm youth (31.84%), whereas the participants from rural desert had 40%. In terms of age, Chi-square analysis indicated that there were significant differences (*χ*^2^_4_ = 15.348, *p* = 0.004) between age groups, with the highest proportion of cluster 1 (unhealthy group) in 15-year-olds (53.01%), and the lowest proportion in 19 years olds (29.60%), whereas the 16-, 17- and 18-year-old groups scored 37.30%, 42.62% and 35.84%, respectively. Moreover, univariate ANOVA was used to identify the differences between the two clusters (i.e., healthy group and unhealthy group) in terms of weight status and sedentary time. The results showed that there was no significant difference in BMI between cluster groups (*p* > 0.05).

There was a significant difference between two clusters in waist circumference (*p* = 0.016), with the healthy group having a lower waist circumference (78.17 cm) than the unhealthy group (80.38 cm). There was also a significant difference between the two clusters in sedentary time (*p* < 0.001). The healthy group spent less time on screen viewing (4.86 h/day) than the unhealthy group (6.07 h/day), as outlined in Table 6, showing the correlation coefficients of BMI and selected lifestyle variables

The current study revealed significant associations between dietary habits and weight status as follows:

#### 3.1.5. Breakfast

Of the sample, 79.96% reported that they did not eat breakfast daily. Breakfast consumption was significantly positively correlated with BMI (*r* = −0.170, *p* < 0.001) and waist circumference (*r* = −0.135, *p* < 0.001). Univariate ANOVA shows that youth who ate breakfast daily had significantly lower BMI (*p* = 0.025) and waist circumference (*p* = 0.001) than those who did not eat breakfast.

#### 3.1.6. Fruit

Of the sample, 87.58% of youth reported that they did not eat fruit daily. However, no significant correlation was found between fruit consumption and BMI and waist circumference (*p* > 0.05).

#### 3.1.7. Vegetables

Of the sample, 77.79% of youth reported, they did not eat vegetables daily. Vegetable consumption had significant negative correlation with waist circumference (*r* = −0.059, *p* = 0.042). However, no significant correlation was found between vegetables consumption and BMI.

#### 3.1.8. Milk and Dairy Product

Of the sample, 74.21% of youth reported that they did not consume dairy products daily. Also, there was no significant correlation found between dairy products consumption, BMI and waist circumference.

#### 3.1.9. Fast Food

Of the sample, 30.47% of youth reported that they ate fast food more than three times per week, but no significant correlation was found between fast food consumption, BMI and waist circumference.

#### 3.1.10. Chips/Crisps

Of the sample, 46.43% of youth reported that they ate chips/crisps more than three time per week. Chips/crisps consumption had a significant negative correlation with BMI (*r* = −0.156, *p* = 0.050). However, no significant correlation was found between chips/crisps consumption and waist circumference (*p* > 0.05).

#### 3.1.11. Soft Drinks

Of the sample, 66.77% of youth reported that they drank soft drinks more than three times per week. No significant correlation was found between soft drink consumption with BMI and waist circumference (*p* > 0.05).

#### 3.1.12. Cakes/Doughnuts

Of the sample, 35.69% of youth reported that they ate cakes/doughnuts more than three times per week. No significant correlation was found between cakes/doughnuts consumption with BMI and waist circumference (*p* > 0.05).

#### 3.1.13. Sweets/Candy

Of the sample, 49.76% of youth reported that they ate sweets/candy more than three times per week. However, no significant correlation was found between sweets/candy consumption with BMI and waist circumference (*p* > 0.05).

#### 3.1.14. Energy Drinks

Of the sample, 15.71% of youth reported that they drank energy drinks more than three times per week. Energy drinks consumption had a highly significant positive correlation with both total BMI (*r* = 0.156, *p* < 0.001) and waist circumference (*r* = 0.156, *p* < 0.001). Moreover, univariate ANOVA revealed that youth who drank energy drinks more than three times per week had significantly higher (*p* < 0.001) BMI and waist circumference.

## 4. Discussion

The aim of this research was to evaluate the nutritional and lifestyle habits of youth in the Al-Ahsa region of KSA. In the last four decades, the KSA has witnessed significant lifestyle transformation and one of the potential contributing factor is the economic boost [11,14]. Subsequently, physical inactivity, sedentary lifestyle and unhealthy dietary habits became predominant especially within the Al-Ahsa region [11,12,14]. The current findings highlight the occurrence of overweight and obesity in youth in the Al-Ahsa Governorate. The present findings align with previous research exploring the association of neighbourhood characteristics and health risk factors among youth in Al-Ahsa Governorate [14], as well as the association of the built environment on hypertension and weight status among adolescence in Al-Ahsa in the KSA [12] and factors influencing the youth of the Al-Ahsa region (PA participation based on the Social Ecological Model) [11]. Furthermore, over one-third of the participants in this study was either overweight or obese. This highlights the importance of health research and culturally tailored interventions being conducted within this region.

The current findings show youth spending an average of (5.34 h) per day on screen time such as watching TV, playing video games, or using a computer and internet (84.79% of the participants (79.18% of males and 90.68% of females)). This excessive screen time is widely recognised as a major factor contributing to obesity [23,24,25]. Furthermore, research reported that the number of hours of screen viewing was associated with the prevalence of having three or more metabolic risk factors [26,27]. The current findings highlight the association between health habits, whilst further longitudinal research is required to explore a potential displacement theory between sedentary time, weight status and PA.

In KSA, dietary patterns have shifted remarkably over the past few decades; the quality of food, location of eating, the amount of food consumption, the availability of restaurants and shops have changed [12]. The current findings highlighted that most participants in this study did not consume breakfast (79.79%), fruit (87.59%), vegetables (77.80), milk and dairy products (74.21%) daily. In addition, there were also high proportions of participants consuming unhealthy foods, such as fast food (30.47%), chips/crisps (46.43%), soft drinks (66.77%), cakes/doughnuts (35.69%), sweets/candy (49.76%) and energy drinks (15.71%), more than three times per week. As noted, dietary habits are coming to the fore as a major modifiable determinant of chronic disease [28,29]. For example, consumption of vegetables and fruit is associated with reduced risk of many chronic diseases [30,31]. Furthermore, Al-Hazzaa, Musaiger [21] found that adolescent Saudi males consumed daily breakfast more than females (71.30% and 79.40%, respectively). Although males scored higher than females in fruit and vegetable consumption, most of them did not consume fruit and vegetables daily, which is below the recommended guideline. Evidence indicates that intake of at least two and half cups of fruit and vegetables per day is associated with a reduced risk of cardiovascular disease, including heart attack, stroke and could protect against the risk of certain types of cancer [32,33,34].

When exploring gender differences, the prevalence of overweight and obesity according to BMI was high in both males (36.6%) and females (39.0%). However, when assessing the abdominal adiposity of youth by measuring waist circumference, significant gender difference was apparent. Waist circumference was measured as a predictor of obesity-related health risk [12,14], and findings revealed that 26.5% of males were classified as at an increased health risk, but more than double the percentage of females were classified as ‘at risk’ (54.2%). The gender differences in the consumption of healthy and unhealthy foods were mixed. However, in terms of the gender sedentary differences, females spend longer than males watching TV and using the internet. This supports previous research conducted in KSA, showing that approximately 84% of youth males and 89% of youth females exceeded two hours per day of screen time [21]. Contrary to the current findings, numerous studies reported that adolescent boys spent more time than girls watching TV or using the computer, or combined [35,36]. This differs from the current findings and might be due to the lack of opportunities for females to be outdoors, as outlined in previous studies [11,12,14].

When exploring the dietary habits of youth, it is important to highlight that the proportion of unhealthy food consumption per day was significantly high among all participants. Whilst males consumed more fast food per day than females (35.12% and 25.54%, respectively), females consumed more than males in chips/crisps (63.48% and 30.39%), cakes/doughnuts (42.18% and 29.61%) and sweets (62.10% and 38.11%). These findings may be due to the differences in societal roles of males and females. For example, females are heavily restricted in where they can travel alone, and this may significantly affect their access to restaurants and fast-food outlets [11,12,14]. Due to such restrictions, females generally spend more of their time within home watching TV or using internet compared to males and subsequently have a greater opportunity to divulge in unhealthy habits that are associated with sedentary time in home, such as cake, sweets and chips consumption. Research reported watching TV for prolonged times could contribute to obesity, including increased SB, displacement of more physical pursuits and unhealthy eating practices such as increased snacking behaviour while viewing and interference with normal sleep patterns [37,38,39].

When the geographical area of the school was considered, the results showed that youth from rural farm had significantly lower BMI and waist circumference, compared to youth from urban and rural desert environments. The BMI scores of rural desert youth were considerably higher than their urban and rural farm counterparts, with (51.2%) of females and (43.5%) of males being classified as overweight or obese. The differences in weight status between youth from different geographical locations could be due to the different PA patterns, including the absence of access to sport and recreational facilities such as parks, playgrounds or swimming pools in the rural desert environment, as outlined in previous studies [11,12,14]. In the Al-Ahsa Governate, the rural desert communities generally have stricter, traditional views on the role and dress code of women, therefore the attitudes, societal norms, and expectations of rural desert communities are less encouraging towards participating in sport and exercise that may require adherence to particular outfits or clothing compared to other communities [11].

The findings are equivocal when comparing dietary habits, with no clear healthy and unhealthy trends or subgroups identified. However, geographical differences were apparent in the sedentary habits. The findings of this study revealed that urban youth spent more time in front of a screen than youth from rural farms and rural desert areas, and that was particularly evident in their use of the internet. This could be due to the fact that urban youth have better access to the internet at internet cafés or at home. Similar findings were suggested in a Chinese study, showing, both boys and girls aged 13–18 years from urban residences spent more time on screen viewing than those in rural areas [40]. A comparative study on SB in American and Canadian youth illustrated that the American youth in the most rural areas were more likely to spend time watching TV and less likely to be high computer users, whereas Canadian youth in metropolitan areas were less likely to spend time watching TV and more likely to be high computer users [41]. In Australia, Dollman, Maher [42] found that young people living in more urban areas spent extra time in front of the screen. However, research documented that people from urban areas have more opportunities to become involved in PA; urban residence has been linked to a sedentary lifestyle due to a lack of adequate space for play, safety concerns, motorised transportation, and computerisation [11,43]. This ambiguity is reflected in studies that investigated PA and SB of urban and rural children, where some studies showed that urban residence fostered a sedentary lifestyle and others found no differences between geographical locations [12,16,44]. These inconsistencies may be due to the different SB measurements. Subsequently, the current study supported some of the previous research findings, indicating that urban youth were more sedentary than rural youth. Further longitudinal and intervention-based research is required to clearly establish what social or environmental factors caused this disparity in sedentary time.

In addition to identifying several differences in the health and lifestyle habits of different demographic groups of Al-Ahsa youth, this study also identified a number of associations between participants’ diet and weight status. For example, regular consumption of breakfast among children has been associated with improved cognitive performance, such as short-term memory, attention, episodic memory and increased learning ability [45,46]. Mechanisms including glucose uptake in the brain as a result of eating breakfast could reflect these cognitive functions [47,48]. It is widely published that not eating breakfast is substantially associated with increased risk of overweight or obesity [49,50,51], based on the reasoning that a complete and well-balanced breakfast keeps away the feelings of hunger, minimising subsequent impulsive unhealthy snacking [52,53]. Furthermore, missing breakfast is associated with lower PA levels and chronic diseases such as type II diabetes [54,55,56,57]. Furthermore, the current findings reflect previous research conducted in KSA and elsewhere, stating that breakfast was the most missed meal among boys and girls across disciplines, ranging from primary to higher education [50,58,59].

The frequency of fruit and vegetable consumption has been linked with respiratory disorder and reducing the risk of cancer in adulthood [60,61]. The current findings demonstrate that (87.58%) male and (77.79%) female youth do not eat fruit and vegetables daily. Furthermore, there is a significant negative correlation between vegetable consumption and waist circumference. Bhupathiraju and Tucker [62] found that the variations in fruit and vegetable consumption were inversely associated with waist circumference, trans-fatty acids, and saturated fatty acids, cardiovascular medications, and diabetes medications. Research also found that children’s weight change was most inversely associated with vegetables, whole grains, fruits, nuts and yogurt consumption [63,64]. Failure to meet the fruit and vegetable guidelines was notably associated with a higher number of unhealthy behaviours [65,66]. In KSA, Amin, Al-Sultan [67] indicated that low servings of fruit and vegetables increase the risk of developing overweight and obesity among school children, while lean students are reported to consume more fruit and vegetables. The present findings support previous research conducted in KSA youth, highlighting the poor dietary habits of young people. Moreover, considering the findings of previous studies within the Middle East and further afield, this emphasises the concerning health risk that youth from the Eastern province of KSA face, with the vast majority not consuming vegetables and fruit daily.

In additon to the substantial associations between the healthy dietary habits and weight status, the results of this study also revealed a number of associations between unhealthy food consumption and lifestyle habits. It was found that (30.47%) of youth eat fast food more than three times per week. However, no significant correlation was found between fast food consumption with BMI and waist circumference. Fast food has higher saturated fat than other foods consumed at home or at school. Due to its poor nutritional quality, fast food is considered a key contributor to increases in the prevalence of overweight and obesity [68,69,70]. Fast food consumption has been associated with other Western dietary patterns such as sugar-sweetened beverages and chips [68,71]. Hence, scholars hypothesised that fast food consumers may have less healthy dietary patterns. Thus, the consumption of fast food might not be directly associated with increased energy intake and weight gain but might instead be a marker for other unhealthy behaviours associated with these outcomes [68]. In support of the previous literature, the current study findings revealed a high proportion of youth who consumed fast food (30.47%), chips/crisps (46.43) and soft drinks (66.77%) more than three times a week. There was a large association between fast food with BMI and waist circumference. However, chips/crisps were inversely associated with BMI but not with waist circumference. The high percentage of Western food consumption is supported by another study, which was conducted in KSA [59], showing that (37.3%) of female university students were eating fast food three or more times a week, (24.5%) of students ate chips/crisps every day and (43.4%) of students consumed soft drinks daily [71]. Research also reported that drinking one or more soft drinks every day increases the prevalence of metabolic syndrome [71].

In the current study, the results showed that (35.69%) of youth consumed sweets/candy and (49.76%) cakes/doughnuts more than three times per week. Moreover, there was no association found between sweets/candy and cakes/doughnuts with weight status. The authors of [67] found sweet and candy consumption frequency increased the risk of developing overweight and obesity among children. Given the high consumption levels of soft drinks revealed in this study, this emphasises the concerning health risks for youth in the Eastern province of KSA.

### Study Strength and Limitations

Whilst highlighting several interesting associations between the health and lifestyle habits of a large representative sample of the Al-Ahsa youth population with different subgroups, this study could not conclude causality in such associations due to its cross-sectional design. Longitudinal research design tracking the health and lifestyle habits of Al-Ahsa youth from adolescence to adulthood would be insightful. The lifestyle habits of youth such as sedentary and dietary have all been assessed using the ATLS questionnaire. Although previous research has used and supported the validity and reliability of ATLS, the limitations of relying upon subjective methods of data collection are well known. Future research adopting a mixed methodological approach and combining objective and subjective measures is recommended.

## 5. Conclusions

This study provided important insight into the current incidence of overweight and obesity, poor dietary habits and SB of youth in the Eastern Province of KSA. To the researchers’ knowledge, this is the first study of its kind in assessing such a broad range of health and lifestyle habits of young males and females in the Al-Ahsa region. Al-Ahsa is the largest governorate in KSA, and, as identified in the study findings, it has some of the largest health risks regarding obesity prevalence and poor lifestyle habits. Such concerning health risks are particularly prevalent in youth from rural desert communities and in females from all demographic backgrounds. The current findings provide a foundation of knowledge for future research to build upon and investigate the major factors that influence the health behaviours of this population. The current findings highlight the major need for research and policy interventions as well as longitudinal studies with culturally tailored programmes to address the concerning health habits of Al-Ahsa youth.

## Figures and Tables

**Table 1 ijerph-22-00813-t001:** Mean ± standard deviation (SD) of the main dependent variables for the total sample and sub-samples.

Variable	Urban		Rural Farm		Rural Desert		Whole Group	
	Male	Female	Male	Female	Male	Female	Male	Female
Age	17.03 ± 1.02	16.78 ± 1.21	17.13 + 1.25	17.50 ± 1.21	17.15 ± 1.04	17.24 ± 1.35	17.08 ± 1.10	17.07 ± 1.27
Weight	70.08 ± 21.62	60.47 ± 20.03	63.31 ± 18.37	54.12 ± 15.51	69.10 ± 20.05	62.23 ± 18.83	67.76 ± 20.62	58.74 ± 18.80
Height	168.6 ± 7.41	154.91 ± 9.00	166.86 ± 6.61	153.75 ± 8.02	167.29 ± 6.07	154.16 ± 5.74	167.86 ± 7.02	154.44 ± 8.32
BMI	24.58 ± 7.18	26.35 ± 21.84	22.67 ± 6.16	23.89 ± 20	24.59 ± 6.66	26.00 ± 6.84	23.97 ± 5.764	25.54 ± 19.83
WC	78.73 ± 16.87	82.47 ± 15.02	73.90 ± 14.15	78.58 ± 12.65	76.36 ± 13.91	81.89 ± 15.24	76.85 ± 15.79	81.17 ± 14.44
Time spent watching TV	2.60 ± 1.81	2.28 ± 1.67	2.29 ± 1.62	2.71 ± 1.97	2.48 ± 2.13	3.63 ± 2.20	2.49 ± 1.80	2.60 ± 1.90
Time spent on computer	2.58 ± 2.02	3.57 ± 2.45	2.34 ± 2.05	2.98 ± 2.29	1.96 ± 2.26	2.12 ± 2.09	2.43 ± 2.07	3.19 ± 2.40

**Table 2 ijerph-22-00813-t002:** The proportions (%) of those who exceeded cut-off scores for dietary habits and sedentary behaviour.

Variables	Males	Females
Breakfast	23.16	16.69 **
Sugary drinks	67.40	66.11
Vegetables	25.04	19.20 *
Fruits	15.43	9.21 *
Milk	29.61	21.74 *
Fast foods	35.12	25.54 **
Crisps/Chips	30.39	63.48 **
Cakes/doughnuts	29.61	42.18 **
Sweets	38.11	62.10 *
Energy Drinks	16.98	14.36
TV	42.57	42.90
Computer	37.87	53.09 **
Sedentary time	79.18	90.68 **

* *p* < 0.05 ** *p* < 0.001.

**Table 3 ijerph-22-00813-t003:** The mean ± SD of the main dependent variables and the proportions (%) of Saudi youth who exceeded certain cut-off values for dietary habits and screen time.

		Urban			Rural Farm			Rural Desert			Whole Groups		
		M	F	Total	M	F	Total	M	F	Total	M	F	Total
Breakfast	Mean and SD (frequency/week)	3.29 ±2.61	2.63 ±2.54	2.98 ±2.60	3.38 ±2.65	2.51 ±2.51	2.96 ±2.62	3.68 ±2.37	3.25 ±2.79	3.47 ±2.60	3.37 ±2.60	2.68 ±2.58	3.03 ±2.61
	% daily intake	22.13	16.01	19.19	24.13	13.90	19.23 a	25.32	25.92	25.63	23.16	16.69	20.04 a
Sugary drinks	Mean and SD (frequency/week)	4.91 ±2.28	4.92 ±2.28	4.92 ±2.28	4.76 ±2.30	4.43 ±2.42	4.60 ±2.36	4.63 ±2.39	5.28 ±2.31	4.96 ±2.37	4.83 ±2.30	4.82 ±2.34	4.82 ±2.32
	% >3 times/week	69.38	68.88	69.14	66.50	57.75	62.31	60.76	74.07	67.5	67.40	66.11	66.77
Vegetables	Mean and SD (frequency/week)	3.84 ±2.50	3.29 ±2.48	3.58 ±2.50	3.90 ±2.28	2.91 ±2.33	3.43 ±2.35	3.41 ±2.44	3.75 ±2.42	3.58 ±2.43	3.81 ±2.42	3.24 ±2.43	3.53 ±2.44
	% daily intake	27.89	20.85	24.49 a	22.89	14.97	19.07 a	17.72	22.22	20	25.04 b	19.20	22.21 a
Fruits	Mean and SD (frequency/week)	3.32 ±2.31	2.65 ±2.11	3.00 ±2.24	3.30 ±2.23	2.31 ±2.13	2.83 ±2.23	2.84 ±2.28	2.94 ±2.19	2.89 ±2.23	3.26 ±2.28	2.58 ±2.14	2.93 ±2.24
	% daily intake	15.21	8.46	11.95 a	16.42	8.11	12.44 a	13.92	14.81	14.38	15.43	9.21	12.42 a
Milk	Mean and SD (frequency/week)	3.99 ±2.48	3.58 ±2.44	3.80 ±2.47	4.15 ±2.48	3.33 ±2.48	3.76 ±2.51	4.15 ±2.62	4.04 ±2.55	4.09 ±2.58	4.06 ±2.49	3.57 ±2.48	3.82 ±2.50
	% daily intake	27.89	21.15	24.64 a	30.85	19.35	25.32 a	34.18	29.63	31.88	29.61	21.74	25.79 a
Fast foods	Mean and SD (frequency/week)	3.42 ±2.02	2.87 ±1.86	3.15 ±1.96	2.84 ±1.96	2.42 ±2.01	2.64 ±1.99	1.79 ±1.94	2.36 ±1.91	2.08 ±1.94	3.03 ±2.06	2.66 ±1.93	2.85 ±2.00
	% >3 times/week	41.69	29.00	35.57 a	31.84	21.93	27.06 a	13.92	19.75	16.88	35.12 b	25.54	30.47 a b
Chips/Crisps	Mean and SD (frequency/week)	2.90 ±2.17	4.57 ±2.26	3.70 ±2.36	2.67 ±2.09	4.43 ±2.40	3.51 ±2.41	1.79 ±2.05	4.83 ±2.56	3.33 ±2.77	2.69 ±2.15	4.56 ±2.34	3.59 ±2.43
	% >3 day/w	32.96	64.35	48.10 a	30.85	62.70	46.11 a	17.72	61.73	40 a	30.39 b	63.48	46.43 a b
Cakes/Doughnuts	Mean and SD (frequency/week)	2.69 ±2.08	3.30 ±2.23	2.98 ±2.17	2.74 ±2.11	3.12 ±2.26	2.92 ±2.19	2.57 ±2.15	4.53 ±2.45	3.56 ±2.50	2.69 ±2.09	3.40 ±2.31	3.03 ±2.23
	% >3 times/week	29.86	40.48	34.99 a	29.35	36.61	32.81	29.11	61.73	45.63 a	29.61	42.18 b	35.69 a b
Sweets	Mean and SD (frequency/week)	3.43 ±2.28	4.63 ±2.17	4.01 ±2.31	2.55 ±2.21	4.08 ±2.35	3.29 ±2.40	2.90 ±2.33	4.72 ±2.24	3.82 ±2.45	3.09 ±2.30	4.47 ±2.25	3.76 ±2.38
	% >3 times/week	44.22	64.95	54.23 a	26.87	55.08	40.46 a	39.24	66.67	53.13 a	38.11 b	62.10	49.76 a b
Energy drinks	Mean and SD (frequency/week)	1.70 ±2.29	1.50 ±2.42	1.60 ±2.35	1.00 ±1.90	0.60 ±1.56	0.81 ±1.75	1.77 ±2.50	1.14 ±2.19	1.45 ±2.36	1.49 ±2.22	1.17 ±2.19	1.33 ±2.21
	% >3 times/week	19.38	19.64	19.51	11.44	5.88	8.76	20.25	12.35	16.25	16.98 b	14.36 b	15.71 b
Sedentary time	Mean and SD (hours/day)	5.18 ±3.03	5.86 ±2.39	5.50 ±2.76	4.63 ±2.76	5.70 ±3.02	5.14 ±2.93	4.43 ±3.37	5.75 ±2.16	5.10 ±2.89	4.91 ±3.00	5.80 ±2.58	5.33 ±2.84
	% >3 times/week	84.13	91.6	87.75 a	75.82	89.41	82.39 a	64.29	89.61	77.55 a	79.18 b	90.68	84.79 a b
	Mean and SD (hours/day)	5.18 ±3.03	5.86 ±2.39	5.50 ±2.76	4.63 ±2.76	5.70 ±3.02	5.14 ±2.93	4.43 ±3.37	5.75 ±2.16	5.10 ±2.89	4.91 ±3.00	5.80 ±2.58	5.33 ±2.84

Data are means and standard deviations (total *n* = 1270). BMI = body mass index. One-way ANOVA tests: a = significant difference between males and females at *p* < 0.05 b = significant gender difference between school types at *p* < 0.05. Contributory risk factors associated with obesity.

**Table 4 ijerph-22-00813-t004:** Odds ratio for risk of increasing obesity (BMI) among males.

	OR	95% CI	*p* Value
TV viewing	1.112	1.011–1.223	0.029
Breakfast	0.926	0.863–0.994	0.032
Dairy products	1.124	1.041–1.214	0.003
Chips and crisps	0.894	0.822–0.974	0.010
Energy drink	1.106	1.021–1.198	0.013

**Table 5 ijerph-22-00813-t005:** Odds ratio for risk of increasing obesity (BMI) among females.

	OR	95% CI	*p* Value
Breakfast	0.922	0.859–0.989	0.023
Energy drink	1.209	1.117–1.308	<0.001

**Table 6 ijerph-22-00813-t006:** Dietary habit association with weight status.

		Breakfast	Sugary Drinks	Vegetables	Fruits	Milk	Fast Foods	Crisp/ Chips	Cakes/ Doughnuts	Sweets	Energy Drinks
BMI	R value	−0.103 **	−0.021	−0.034	0.000	0.042	0.042	−0.056 *	0.018	−0.031	0.156 **
	*p* value	0.000	0.473	0.237	0.993	0.140	0.142	0.050	0.542	0.288	0.000
TV Viewing per day	R value	0.018	0.102 **	0.031	0.033	0.017	0.062 *	0.100 **	0.128 **	0.105 **	0.018
	*p* value	0.531	0.000	0.269	0.245	0.541	0.029	0.000	0.000	0.000	0.538
Computer time per day	R value	−0.083 **	0.100 **	−0.062 *	−0.081 **	−0.078 **	0.205 **	0.207 **	0.099 **	0.180 **	0.139 **
	*p* value	0.004	0.000	0.029	0.004	0.006	0.000	0.000	0.001	0.000	0.000
sedentary time	R value	−0.054	0.147 **	−0.029	−0.044	−0.051	0.204 **	0.231 **	0.163 **	0.212 **	0.123 **
	*p* value	0.056	0.000	0.307	0.127	0.076	0.000	0.000	0.000	0.000	0.000

* Correlation is significant at the 0.05 level (2-tailed). ** Correlation is significant at the 0.01 level (2-tailed).

## Data Availability

Data are contained within the article, but the raw data can be requested with a reasonable request from the authors.

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
