# Peer review of "Exploring Sedentary and Nutritional Behaviour Patterns in Relation to Overweight and Obesity Among Youth from Different Demographic Backgrounds in Saudi Arabia"

_ijerph, 2025, doi:10.3390/ijerph22050813_

Round 1
Reviewer 1 Report
Comments and Suggestions for Authors
Comments to the Author
This randomized cross-sectional study was to explore the nutritional and lifestyle habits of youth in Al-Ahsa region of KSA. The findings reveal the high prevalence of overweight and obesity in youth in the Al-Ahsa governorate. The paper is commendably written and organized. Findings in this research emphasize the importance of health research and interventions being conducted within this region. However, some aspects of the methodology and discussion sections may benefit from further clarification and revision. Specific comments follow.
Introduction:
- Line 76-91: A more detailed sampling method is needed. For a random stratified sampling method based on different regions, genders, and age groups, is the number of samples evenly distributed? How is it implemented?
- Base on Line 109-112: The Arab Teens Lifestyle Survey (ATLS) was developed 15 years ago and is relatively outdated for investigating the current lifestyles of today's youth. Has the ATLS questionnaire been modified in this study? If so, the revisions should be clearly specified. If not, please explain this as a limitation in the study.
- The table is unclear, which makes it unfriendly to read.
- Tables 1 and 2 should be presented in sequential order.
- There are two Table 2 in this manuscript.
- The data presented in the tables should be rounded to one decimal place for consistency.
- Line 181: Table 2. The proportions (%) of who exceeded cut-off scores, please show the cut-off scores on the footnote.
- Table 2. The letters a and b in the table should be formatted as superscripts.
- Line 322: Please add Table number
- Line 279: females were more likely to be overweight by 18% (OR=.922; 95% CI.859.989). why 18%? I don’t understand.
- Line 328-329: Breakfast consumption was significantly negatively correlated with BMI (r= −0.170, p< 0.001) which is different to correlation Table?
- I appreciate your discussion section — the arguments are well-structured and supported by sufficient literature. A few minor issues are as follows:
- From Tables 3 and 4, as well as the correlation table, it is evident that energy drink consumption has a positive relationship with BMI, sedentary time, and daily computer usage (R value). This is a significant finding; however, the authors did not address it in the discussion section.
- Line 50-51: Al-Nuaim and Safi 2022) found boys were spending more time being sedentary (3.14) compared to girls (2.96) hours per day sitting and significant difference was found. It seems differ compare to your study (Table 2). Why?
Author Response
Dr Ayazullah Safi
Birmingham City University
Westbourne road,
Edgbaston
B15 3TN
24th April 2025
Dear Reviewer 1,
International Journal of Environmental Research and Public Health
Thank you for reviewing our manuscript which is prepared for the above journal.
We are pleased to have completed necessary amendments and have uploaded this new version online as requested. I also attach responses to your comments in the table below.
Please let me know if I can provide you with any further information. Thank you for your time.
Yours sincerely,
Ayazullah Safi
COMMENT NO. |
REVIEWER 1 COMMENTS
|
Authors RESPONSE
|
|
Brief Summary
|
|
1 |
Comments to the Author
|
Thank you for this observation. We have made the necessary amendments as requested. |
2 |
Line 76-91: A more detailed sampling method is needed. For a random stratified sampling method based on different regions, genders, and age groups, is the number of samples evenly distributed? How is it implemented?
|
Thank you for this observation. The information about the sample size including age, gender and geographical location have been explained and further clarified in line 76-91.
|
3 |
Base on Line 109-112: The Arab Teens Lifestyle Survey (ATLS) was developed 15 years ago and is relatively outdated for investigating the current lifestyles of today's youth. Has the ATLS questionnaire been modified in this study? If so, the revisions should be clearly specified. If not, please explain this as a limitation in the study.
|
This has added to the limitation in line 570-572. |
4 |
The table is unclear, which makes it unfriendly to read. Tables 1 and 2 should be presented in sequential order. There are two Table 2 in this manuscript. Line 181: Table 2. The proportions (%) of who exceeded cut-off scores, please show the cut-off scores on the footnote. Table 2. The letters a and b in the table should be formatted as superscripts. Line 322: Please add Table number.
|
Thank you for this observation. These points have been revised and addressed for clarity accordingly. |
|
Line 328-329: Breakfast consumption was significantly negatively correlated with BMI (r= −0.170, p< 0.001) which is different to correlation Table?
|
Thank you for this observation. Good observation this has now been addressed. |
|
I appreciate your discussion section — the arguments are well-structured and supported by sufficient literature. A few minor issues are as follows: From Tables 3 and 4, as well as the correlation table, it is evident that energy drink consumption has a positive relationship with BMI, sedentary time, and daily computer usage (R value). This is a significant finding; however, the authors did not address it in the discussion section.
|
Thank you for this observation. We tried and added more details to the discussion and hope this query is now addressed. |
|
Line 50-51: Al-Nuaim and Safi 2022) found boys were spending more time being sedentary (3.14) compared to girls (2.96) hours per day sitting and significant difference was found. It seems differ compared to your study (Table 2). Why?
|
Thanks for this but not sure if we need to address this as the study by Al-Nuaim and Safi 2022 was looking at the sedentary time as well as physical activity levels which suggested that even though boys were more active, but they were more sedentary, and this has been addressed in the original study of Al-Nuaim and Safi 2022. Is there any specific point you want us to address in current study? |
Reviewer 2 Report
Comments and Suggestions for Authors
Dear authors,
Thank you for the opportunity. While the study covers an important and under-researched area and provides valuable insights, it requires revisions for clarity, coherence, and precision in language and analysis. See the specific feedback for each section.
*Abstract*
The abstract effectively summarizes the study objectives, methodology, key findings, and conclusions. However, it contains grammatical errors (e.g., "Given the overwhelming problem that obesity poses to Kingdome of Saudi Arabia") and can be improved for fluency. Consider streamlining the abstract by avoiding redundancy (e.g., “poor dietary habits and sedentary behaviour” is repeated in the conclusion line unnecessarily).
*Introduction / Background*
Significant repetition exists across paragraphs, especially when discussing the shift in lifestyle and dietary patterns in KSA. Several grammatical and typographical issues detract from clarity (e.g., “condcuted”, “shows that obesity amongst youth…” should be “show”). Overall, the ntroduction lacks a clear research question or hypothesis; it should explicitly state what is novel about this study compared to existing work.
*Methods*
Ethical considerations are mentioned briefly; a more detailed description of the ethical process, including approval code, would improve transparency. The section on statistical analysis is slightly disorganized. Consider separating descriptive and inferential methods more clearly. Clarify whether any missing data were present and how they were handled. *Results* This section is good. The results are comprehensive, using tables and statistical tests effectively. The stratification by gender, age, location, and school type adds depth to the analysis.
*Discussion*
Connects the findings to the broader literature and contextualizes them within local cultural and infrastructural realities. However, provide a stronger justification or discussion of the limitations.
*Conclusion*
Try to be more concise, a focused restatement of key findings and their implications. Include a more direct statement of what future research should explore (e.g., longitudinal effects, intervention strategies).
Author Response
Dr Ayazullah Safi
Birmingham City University
Westbourne road,
Edgbaston
B15 3TN
24th April 2025
Dear Reviewer 2,
International Journal of Environmental Research and Public Health
Thank you for reviewing our manuscript which is prepared for the above journal.
We are pleased to have completed necessary amendments and have uploaded this new version online as requested. I also attach responses to your comments in the table below.
Please let me know if I can provide you with any further information. Thank you for your time.
Yours sincerely,
Ayazullah Safi
COMMENT NO. |
REVIEWER 1 COMMENTS
|
Authors RESPONSE
|
|
Brief Summary
|
|
1 |
Thank you for the opportunity. While the study covers an important and under-researched area and provides valuable insights, it requires revisions for clarity, coherence, and precision in language and analysis. See the specific feedback for each section. |
Thank you for reviewing our manuscript. We have made the necessary amendments as requested. |
2 |
The abstract effectively summarizes the study objectives, methodology, key findings, and conclusions. However, it contains grammatical errors (e.g., "Given the overwhelming problem that obesity poses to Kingdome of Saudi Arabia") and can be improved for fluency. Consider streamlining the abstract by avoiding redundancy (e.g., “poor dietary habits and sedentary behaviour” is repeated in the conclusion line unnecessarily). |
Thank you for this observation. This is clarified and re-written hope it is clear now.
|
3 |
Significant repetition exists across paragraphs, especially when discussing the shift in lifestyle and dietary patterns in KSA. Several grammatical and typographical issues detract from clarity (e.g., “condcuted”, “shows that obesity amongst youth…” should be “show”). Overall, the introduction lacks a clear research question or hypothesis; it should explicitly state what is novel about this study compared to existing work. |
Thank you for this observation. We have revised the introduction and proofread for clarity. We hope it is clear now and happy to make further revisions as required. |
4 |
Ethical considerations are mentioned briefly; a more detailed description of the ethical process, including approval code, would improve transparency. The section on statistical analysis is slightly disorganized. Consider separating descriptive and inferential methods more clearly. Clarify whether any missing data were present and how they were handled. *Results* This section is good. The results are comprehensive, using tables and statistical tests effectively. The stratification by gender, age, location, and school type adds depth to the analysis. Thank you, this has been addressed and revised. Also, results section has been revised for further clarity.
|
Thank you for this observation. Further details about ethical approval including reference/code have been added. Also, amendments to methods as well as results were condcuted for clarity. Happy to make further adjustment as required. |
|
Connects the findings to the broader literature and contextualizes them within local cultural and infrastructural realities. However, provide a stronger justification or discussion of the limitations. |
Thank you for this observation. Revisions made and adjusted as requested. Happy to make further adjustment as required. |
|
Try to be more concise, a focused restatement of key findings and their implications. Include a more direct statement of what future research should explore (e.g., longitudinal effects, intervention strategies). This has been revised
|
Thank you for this observation. Revisions made and adjusted as requested. Happy to make further adjustment as required. |
Reviewer 3 Report
Comments and Suggestions for Authors
Review of the article on “Exploring the sedentary and nutritional behaviour patterns of youth from different demographic backgrounds in Saudi Arabia”.
Formal objections
The article does not meet the requirements for publication:
- references are not cited in the order of appearance, with numbers given in brackets
- identification data of the bioethics committee consent have not been disclosed
- no statements have been included: Author Contributions, Funding, Institutional Review Board Statement, Informed Consent Statement, Data Availability Statement, Conflicts of Interest, etc.
- suggested to include a list of abbreviations
Linguistic objections
Numerous incorrect or unclear linguistic entries, e.g.:
- “Given the overwhelming problem that obesity poses to Kingdome of Saudi Arabia (KSA) population.” (lines 11-12)
- “Some of the key contributing factors reported to be associated with behavioural, poor diet, sedentary lifestyle, expansion of urbanisation, modes of transportation, availability of entertainment gadgets, and lack of physical activity (PA) engagement (Al-Nuaim and Safi, 2023; Badawi and Farag, 2021; Boakye et al., 2023).” (lines 44-47)
- “Furthermore, Al-Nuaim and Safi (2022) conducted a study focused on the correla-48 tion of built environment on hypertension and weights status (…)” (lines 48-49), etc.
Substantive doubts
- the study concerns overweight and obesity, hence the proposal to change the title and develop the results so that they indicate factors responsible for overweight and obesity or body mass
- when discussing the results of the study, do not cite the statistical methods used as tools, but provide (only) the p value
- the introduction does not contain a research hypothesis
- no explanations are provided regarding the cut-off points for data on the frequency of consumption, lifestyle (recommendations?, what?)
- did the obtained data have parametric distributions?
- I suggest using the regression method to identify factors that are important for the body mass of the study participants, including in different environments and for the entire group
- the description of the results requires shortening, indicating only the significance (without describing the detailed data from the tables)
- the discussion requires clear shortening with division into sections according to the previously discussed results
Linguistic objections
Numerous incorrect or unclear linguistic entries, e.g.:
- “Given the overwhelming problem that obesity poses to Kingdome of Saudi Arabia (KSA) population.” (lines 11-12)
- “Some of the key contributing factors reported to be associated with behavioural, poor diet, sedentary lifestyle, expansion of urbanisation, modes of transportation, availability of entertainment gadgets, and lack of physical activity (PA) engagement (Al-Nuaim and Safi, 2023; Badawi and Farag, 2021; Boakye et al., 2023).” (lines 44-47)
- “Furthermore, Al-Nuaim and Safi (2022) conducted a study focused on the correla-48 tion of built environment on hypertension and weights status (…)” (lines 48-49), etc.
Author Response
Dr Ayazullah Safi
Birmingham City University
Westbourne road,
Edgbaston
B15 3TN
24th April 2025
Dear Reviewer 3,
International Journal of Environmental Research and Public Health
Thank you for reviewing our manuscript which is prepared for the above journal.
We are pleased to have completed necessary amendments and have uploaded this new version online as requested. I also attach responses to your comments in the table below.
Please let me know if I can provide you with any further information. Thank you for your time.
Yours sincerely,
Ayazullah Safi
COMMENT NO. |
REVIEWER 1 COMMENTS
|
Authors RESPONSE
|
|
Brief Summary
|
|
1 |
Review of the article on “Exploring the sedentary and nutritional behaviour patterns of youth from different demographic backgrounds in Saudi Arabia”.
|
Thank you for reviewing our manuscript |
2 |
The article does not meet the requirements for publication:
|
Thank you for this observation. Significant changes (e.g., revisions) are made to the manuscript as per your and other reviewers’ suggestions.
|
3 |
references are not cited in the order of appearance, with numbers given in brackets
|
This has been addressed, and we are happy to make further adjustment as required subject to the outcome of decision. |
4 |
identification data of the bioethics committee consent have not been disclosed
|
Thank you for this observation. We have added more details to address this query. |
|
- no statements have been included: Author Contributions, Funding, Institutional Review Board Statement, Informed Consent Statement, Data Availability Statement, Conflicts of Interest, etc.
|
Thank you for this observation. Good observation this has now been added and addressed. |
|
- suggested to include a list of abbreviations
|
Thank you for this observation. Abbreviation has been added throughout the manuscript |
|
Linguistic objections Numerous incorrect or unclear linguistic entries, e.g.:
|
Thanks for this observation. We have made significant revisions to the manuscript which will address these queries and add further clarity. |
|
the study concerns overweight and obesity, hence the proposal to change the title and develop the results so that they indicate factors responsible for overweight and obesity or body mass
|
Thank you for this observation. We are not clear on what exactly this comment is asking us to do. Happy to adjust accordingly if required and if clarified please. |
|
- when discussing the results of the study, do not cite the statistical methods used as tools, but provide (only) the p value
|
Thank you for this observation. Given the various statistical tests, we conducted we believe that adding the statistical methods may add clarity for the reader and find it easy to understand the results. |
|
the introduction does not contain a research hypothesis
|
Thank you for this observation. The introduction contains a clear aim which we believe is sufficient for this manuscript. |
|
did the obtained data have parametric distributions?
|
Thank you for this observation. We believe sufficient details about the data analysis is provided. |
|
I suggest using the regression method to identify factors that are important for the body mass of the study participants, including in different environments and for the entire group
|
Thank you for this observation. We believe sufficient analysis is conducted. |
|
the description of the results requires shortening, indicating only the significance (without describing the detailed data from the tables)
|
Thank you for this observation. Although, we made some changes and revisions as per other reviewers. However, given the needed research we believe the results are sufficiently presented. |
|
the discussion requires clear shortening with division into sections according to the previously discussed results
|
Thank you for this observation. Revision and adjustment have been made to the discussion section. Also, a breakdown into sections considering existing literature has been provided. |
Round 2
Reviewer 3 Report
Comments and Suggestions for Authors
Review of the article on “Exploring the sedentary and nutritional behaviour patterns of youth from different demographic backgrounds in Saudi Arabia”.
Formal objections
The article does not meet the requirements for publication:
- references should be cited in order of appearance, with numbers in parentheses
Linguistic objections
There are still many incorrect or unclear linguistic entries, e.g.:
- “The prevalence of overweight and obesity has increased over the last three decades becoming a major public health concern particularly due to the significant impact obesity has on the population to Kingdome of Saudi Arabia (KSA).” (lines 10-13)
- “Some of the key contributing factors reported including poor diet, sedentary lifestyle, expansion of urbanisation, modes of transportation, availability of entertainment gadgets, and lack of physical activity (PA) engagement” (lines 46-48)
- “Furthermore, Al-Nuaim and Safi (2022) conducted a study focused on the correlation of built environment on hypertension and weights status amongst youth in KSA, result showed, boys were spending more time being sedentary (3.14) compared to girls (2.96) hours per day sitting and significant difference was found (F1,379 = 16.50, p < 0.001) between boys (mean = 75.80 cm) and girls (mean = 70.38 cm) waist circumference.” (lines 49-54), etc.
Substantive doubts - I still maintain that:
- the study concerns overweight and obesity, hence the proposal to change the title and develop the results so that they indicate the factors responsible for overweight and obesity or body mass
- discussing research results, I suggest not giving the names of statistical methods as a tool, but the p value (statistical significance) or the percentage of changes for the result and statistical significance
- the Authors changed the hypothesis to “Given the significant impact of obesity on the population of the KSA, it is important to focus on the relatively under-researched health and lifestyles behaviors of young people in the country. Therefore, the aim of this research was to explore the nutritional and lifestyle habits of youth in the Al-Ahsa region of KSA.”, which is still not consistent with the title without taking into account the context of body weight.
- no explanations are provided regarding the cut-off points for data on the frequency of consumption, lifestyle (recommendations?, what?)
- did the obtained data have parametric distributions?
- I suggest using the regression method to identify factors that are important for the body mass of the study participants, including in different environments and for the entire group
- the description of the results requires shortening, indicating only the significance (without describing the detailed data from the tables)
- the discussion requires clear shortening with division into sections according to the previously discussed results
Comments on the Quality of English LanguageLinguistic objections
There are still many incorrect or unclear linguistic entries, e.g.:
- “The prevalence of overweight and obesity has increased over the last three decades becoming a major public health concern particularly due to the significant impact obesity has on the population to Kingdome of Saudi Arabia (KSA).” (lines 10-13)
- “Some of the key contributing factors reported including poor diet, sedentary lifestyle, expansion of urbanisation, modes of transportation, availability of entertainment gadgets, and lack of physical activity (PA) engagement” (lines 46-48)
- “Furthermore, Al-Nuaim and Safi (2022) conducted a study focused on the correlation of built environment on hypertension and weights status amongst youth in KSA, result showed, boys were spending more time being sedentary (3.14) compared to girls (2.96) hours per day sitting and significant difference was found (F1,379 = 16.50, p < 0.001) between boys (mean = 75.80 cm) and girls (mean = 70.38 cm) waist circumference.” (lines 49-54), etc.
Author Response
Dr Ayazullah Safi
Birmingham City University
Westbourne road,
Edgbaston
B15 3TN
09th May 2025
Dear Reviewe,
International Journal of Environmental Research and Public Health
Thank you for reviewing our manuscript which is prepared for the above journal.
We are pleased to have completed necessary amendments and have uploaded this new version online as requested. I also attach responses to your comments in the table below.
Please let me know if I can provide you with any further information. Thank you for your time.
Yours sincerely,
Ayazullah Safi
COMMENT NO. |
REVIEWER 1 COMMENTS
|
Authors RESPONSE
|
|
Brief Summary
|
|
1 |
References should be cited in order of appearance, with numbers in parentheses |
Thank you for this observation. References are now provided as numbered. If needed further adjustment will be made in due course. |
2 |
The prevalence of overweight and obesity has increased over the last three decades becoming a major public health concern particularly due to the significant impact obesity has on the population to Kingdome of Saudi Arabia (KSA).” (lines 10-13) |
Thank you for this observation. This is now addressed for clarity.
|
3 |
“Some of the key contributing factors reported including poor diet, sedentary lifestyle, expansion of urbanisation, modes of transportation, availability of entertainment gadgets, and lack of physical activity (PA) engagement” (lines 46-48) |
Thank you for this observation. This is now addressed for clarity.
|
4 |
Furthermore, Al-Nuaim and Safi (2022) conducted a study focused on the correlation of built environment on hypertension and weights status amongst youth in KSA, result showed, boys were spending more time being sedentary (3.14) compared to girls (2.96) hours per day sitting and significant difference was found (F1,379 = 16.50, p < 0.001) between boys (mean = 75.80 cm) and girls (mean = 70.38 cm) waist circumference.” (lines 49-54), etc.
|
Thank you for this observation. This is now addressed for clarity. . |
|
the study concerns overweight and obesity, hence the proposal to change the title and develop the results so that they indicate the factors responsible for overweight and obesity or body mass |
Thank you for this observation. Element of overweight and obesity is now added to the title. |
|
discussing research results, I suggest not giving the names of statistical methods as a tool, but the p value (statistical significance) or the percentage of changes for the result and statistical significance
|
Thank you for this observation. Given the various statistical tests, we conducted we believe that adding the statistical methods may add clarity for the reader and find it easy to understand the results. |
|
the Authors changed the hypothesis to “Given the significant impact of obesity on the population of the KSA, it is important to focus on the relatively under-researched health and lifestyles behaviors of young people in the country. Therefore, the aim of this research was to explore the nutritional and lifestyle habits of youth in the Al-Ahsa region of KSA.”, which is still not consistent with the title without taking into account the context of body weight. |
Thanks for this observation. As element of overweight and obesity is now added to the title – we hope this satisfy this query. |
|
the study concerns overweight and obesity, hence the proposal to change the title and develop the results so that they indicate factors responsible for overweight and obesity or body mass
|
Thank you for this observation. We are not clear on what exactly this comment is asking us to do. However, element of overweight and obesity is added to the title and the results are clearly provided and discussed in respective sections. |
|
no explanations are provided regarding the cut-off points for data on the frequency of consumption, lifestyle (recommendations? what?)
|
Thank you for this observation – cut off points are not added to the measures section line 99 |
|
did the obtained data have parametric distributions?
|
Thank you for this observation. The data was approximately normally distributed, satisfying the assumption of normality required for parametric analysis. |
|
I suggest using the regression method to identify factors that are important for the body mass of the study participants, including in different environments and for the entire group |
Thank you for this observation. We believe sufficient analysis has been conducted already including the cluster, using 2-way and 3-way analyses of variance (ANOVA) - Chi-square and analysis of variance (ANOVA) were used. |
|
the description of the results requires shortening, indicating only the significance (without describing the detailed data from the tables) |
Thank you for this observation. Although, we made some changes and revisions as per other reviewers. However, given the needed research we believe the results are sufficiently presented. |
|
the discussion requires clear shortening with division into sections according to the previously discussed results
|
Thank you for this observation. We believe that discussion section is sufficiently presented as per the previous round comments and amendments. |
